# Performance and Stability of Wet-Milled CoAl₂O₄, Ni/CoAl₂O₄, and Pt,Ni/CoAl₂O₄ for Soot Combustion

**Carmen M. Álvarez-Docio [1],\* [ID], Raquel Portela [2],\* [ID], Julián J. Reinosa [1] [ID],
Fernando Rubio-Marcos [1,3] [ID], Laura Pascual [2] and José F. Fernández [1] [ID]**

[1]   Instituto de Cerámica y Vidrio, Consejo Superior de Investigaciones Científicas (CSIC), Kelsen 5,
      28049 Madrid, Spain; jjreinosa@icv.csic.es (J.J.R.); frmarcos@icv.csic.es (F.R.-M.); jfernandez@icv.csic.es (J.F.F.)
[2]   Instituto de Catálisis y Petroleoquímica, Consejo Superior de Investigaciones Científicas (CSIC),
      Marie Curie 2, 28049 Madrid, Spain; Laura.pascual@icp.csic.es
[3]   Escuela Politécnica Superior, Universidad Antonio de Nebrija, C/Pirineos, 55, 28040 Madrid, Spain
\*   Correspondence: carmenma.docio@icv.csic.es (C.M.Á.-D.); raquel.portela@csic.es (R.P.);
      Tel.: +34-917355840 (C.M.Á.-D.); +34-915854873 (R.P.)

**Abstract:** Low-energy wet milling was employed to activate commercial CoAl₂O₄ spinel and disperse mono- and multimetallic nanoparticles on its surface. This method yielded efficient Pt,Ni catalysts for soot oxidation in simulated diesel exhaust conditions. The characterization and activity results indicated that although Ni/CoAl₂O₄ was highly active, the presence of Pt was required to obtain a stable Ni(0.25 wt. %),Pt(0.75 wt. %)/CoAl₂O₄ catalyst under the operating conditions of diesel particulate filters, and that hot spots formation must be controlled to avoid the deactivation of the cobalt aluminate. Our work provides important insight for new design strategies to develop high-efficiency low-cost catalysts. Platinum-containing multimetallic nanostructures could efficiently reduce the amount of the costly, but to date non-replaceable, Pt noble metal for a large number of industrially important catalytic processes.

**Keywords:** diesel soot oxidation; CoAl₂O₄ spinel; platinum-based catalyst; nanodispersion

## 1. Introduction

Diesel internal-combustion engine emissions could rise between 50% and 250% by 2050 [1] despite the increasingly stringent regulations for vehicles in both the EU and the US, and, therefore, even more regulation in this sector is expected to limit their impact on health and the environment [2]. The pressure to comply with the norms has led to misbehaviors, such as the "dieselgate" in the automotive industry [3], and efforts are continuously made to further control and reduce diesel emissions of NO$_x$ and particulate matter, the latter comprising soot and other smaller molecular compounds, mostly toxic [4–6].

State-of-the-art diesel particulate filters have considerably reduced soot emissions by trapping the carbonaceous particles inside the filter pores and their subsequent oxidation to CO₂ [7]. Un-catalyzed soot combustion occurs at around 600 °C, and, therefore, a catalyst is added to reduce this temperature below that of diesel exhausts, in the range from 260 to 540 °C [1,8,9]. Noble metals, such as Pt, achieve high catalytic activity by oxidizing the NO present in the exhaust gas to NO₂, which promotes soot combustion [10,11]. The remaining challenges are to avoid filter thermal regeneration and to minimize the amount of scarce and costly Pt [12–14]. This drives the quest for improving the activity and reducing the noble metal loading [15–17]. Furthermore, promotion by alkali metals is reported to increase the catalyst ability to form compounds with low melting point temperatures [18]. This is meant to increase the number of contact points between the catalyst surface and soot particles, which is essential for soot oxidation [19–22]. In this context, mixed oxides built of d-metal cations (Fe, Mn, Co), which

work as a redox center, and an alkali cation, most often potassium, have been investigated [23,24]. However, the stability of potassium is poor, since evaporation leads to a loss of catalyst mass at high temperatures, which restricts the actual application of potassium-containing catalysts [25]. In this regard, other synthetic strategies have been proposed, including the control over particles' shape and size [11,26], the generation of Pt-based materials with hollow interiors [27], and bi- and trimetallic compositions [28,29]. Multimetallic catalysts have the advantage of tunable electronic and geometric properties through different combinations of metal components [30–33]. In this sense, the partial substitution of platinum with nickel to form a bimetallic Pt,Ni catalyst may lead to optimized properties for soot combustion with $NO/O_2$ while reducing the metal content/price. Ni dispersed on various supporting materials ($Al_2O_3$, $CeO_2$, $ZrO_2$, $SiO_2$, $TiO_2$) has been extensively used in hydrogenation, hydro-treating, $CO_2$ methanation [34,35], and steam-reforming reactions due to its good performance and low cost [36]. Moreover, compared to monometallic Ni and Pt, bimetallic Pt,Ni catalysts exhibit higher activity for several reactions [37].

The activity and stability of the active phases depend on the material used as support [38–40], which not only helps in dispersing the active component, but may also participate in some steps of the reaction mechanism [41]. Therefore, supports play an important role in heterogeneous catalysis, especially with metals, and must also be optimized [42]. $Al_2O_3$ is widely used as support due to its high surface area and thermal stability. However, $Al_2O_3$ is inert itself for both NO and soot oxidation [43]. Some highly active basic supports like ceria easily suffer from sulfur-poisoning due to their basicity [44,45]. Hence, the use of appropriate acidic supports may give rise to more active and durable catalysts for soot oxidation. Mixed oxides with spinel structure may be advantageous to support metal catalysts. For instance, Pt nanoparticles on the {111} facets of a $MgAl_2O_4$ spinel have shown great dispersion and stability in harsh conditions [46], and several cobalt spinels efficiently catalyze soot combustion through NO oxidation promotion [47–49]. We have recently reported that the creation of acid sites and defects in commercially available $CoAl_2O_4$ by wet milling yields a low-cost active catalyst for soot oxidation [50]. This material could be used as support where the oxygen vacancies serve as strong anchoring sites for metal nanoparticles (NPs) [51]. Moreover, the metal dispersion could also be performed mechanically in the mill used for the support activation, as this method has been used to prepare a $Pt/Al_2O_3$ catalyst with very good results [52].

Taking this background into account, the aim of the current work was to disperse Ni and Ni-Pt nanoparticles by low-energy milling onto activated cobalt aluminate spinel to obtain promoted catalytic activity at a reduced cost. The performance and stability of the $Pt,Ni/CoAl_2O_4$ and $Ni/CoAl_2O_4$ catalysts for soot oxidation were evaluated. As far as we know, the use of $CoAl_2O_4$-supported platinum and/or nickel catalysts has not been previously reported.

## 2. Results and Discussion

### 2.1. Textural and Structural Characterization

The morphology of the materials was appreciated in the SEM micrographs of Figures 1 and 2. Both $Ni/CoAl_2O_4$ and $Pt,Ni/CoAl_2O_4$ catalysts presented agglomerates, <10 μm in size (a), formed by metal nanoparticles of ca. 50 nm anchored onto submicronic $CoAl_2O_3$ particles with irregular morphology and average size ca. 500 nm (b). The nanoparticles seemed to be strongly bonded, as reported for other materials prepared by similar dispersion methods [53,54]. The EDX mappings included indicated that in both nanostructured samples the metal nanoparticles were uniformly distributed throughout the agglomerates.

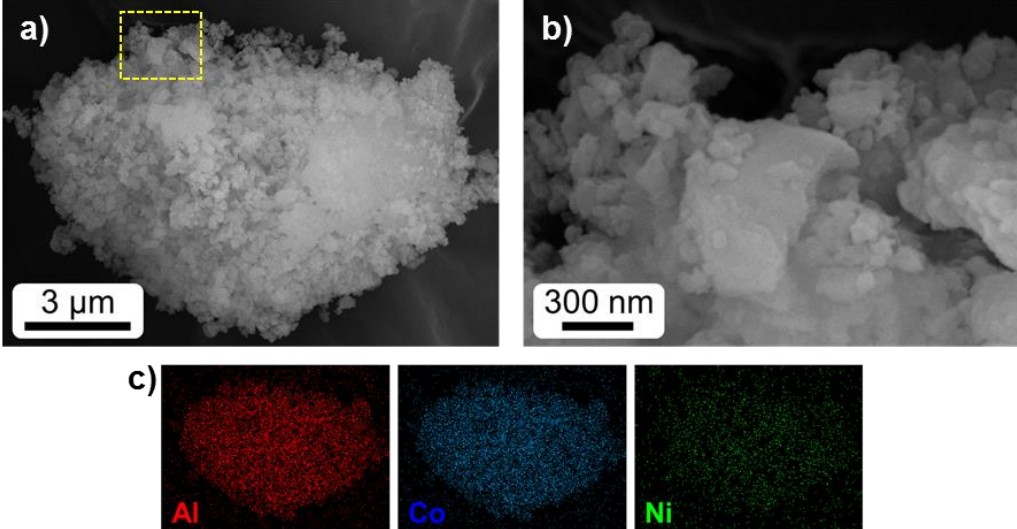

**Figure 1.** (**a**) FE-SEM micrograph of the Ni/CoAl$_2$O$_4$ catalyst, (**b**) HR-SEM magnification of the area marked in yellow in (**a**), and (**c**) EDX elemental mapping obtained in (**a**).

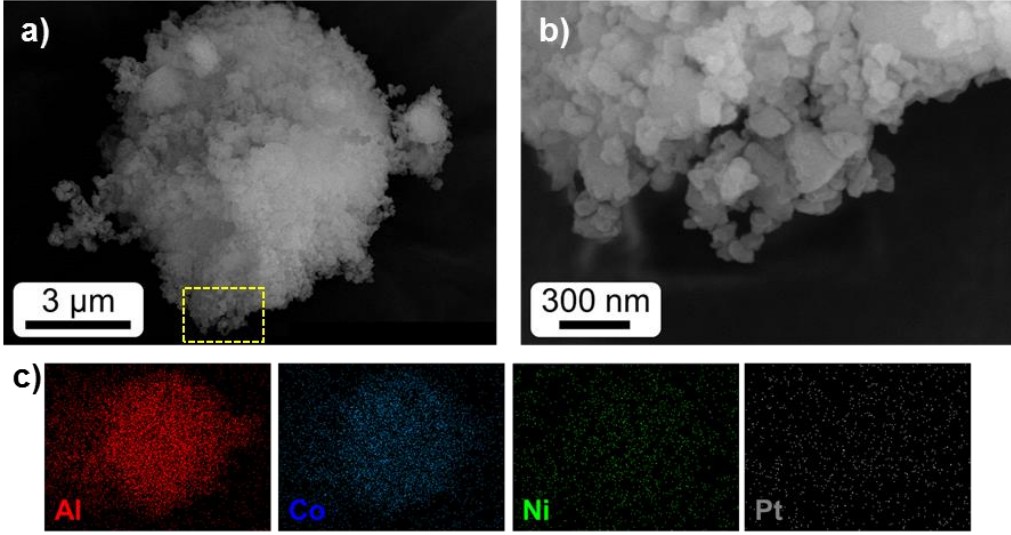

**Figure 2.** (**a**) FE-SEM micrograph of the Pt,Ni/CoAl$_2$O$_4$ catalyst, (**b**) HR-SEM magnification of the area marked in yellow in (**a**), and (**c**) EDX elemental mapping obtained in (**a**).

TEM provided more details on the microstructure of the samples. Figure 3 shows representative micrographs of the Pt,Ni/CoAl$_2$O$_4$ catalyst, confirming that dispersed spherical metal nanoparticles were anchored to CoAl$_2$O$_4$ particles of irregular morphology and different sizes (Figure 3a). The dispersed nano-sized Pt and Ni particles were uniformly distributed, either as agglomerates (>20 nm) or as small nanoparticles (~3 nm) (Figure 3b). The SAED (Selected Area Electron Diffraction) pattern and EDX analyses, corresponding to the area of Figure 3b, are shown in Figure 3c,d, respectively. The diffraction rings, in which the four first reflections are been marked, could be ascribed to the CoAl$_2$O$_4$ spinel structure (spatial group Fd3m). The EDX analysis detected the presence of Pt and Ni, together with Al and Co; however in the SAED pattern, no diffraction spots could be indexed as Pt or Ni phases because of their small size and quantity. Such conclusions were drawn from the results of XRD studies (Figure S1). XPS analyses of the fresh and used samples revealed traces of zirconia contamination, attributed to the milling procedure [55], and indicated that all the samples had a normal spinel structure and Co$^{2+}$ cations in tetrahedral sites (Figure S2, Table S1) [56,57]. Figure S3a shows a STEM-HAADF (Scanning Transmission Electron Microscopy-High Angle Annular Dark Field)

illustrative image of a region with bright spots dispersed all over the $CoAl_2O_4$ that correspond to Pt and Ni, according to the EDX mapping of these elements in the region. The semi-quantitative distribution of the metals along the $CoAl_2O_4$ support determined by EDX was 65% Pt and 35% Ni, in the range of the theoretical catalyst formulation. In the STEM-HAADF of another area presented in Figure 3e, two nanoparticles of Ni and Pt have been analyzed in detail to understand their nanostructure. In the case of Ni (Figure 3f), the FFT (Fast Fourier Transform) of the HRTEM image could be indexed as an (FM3-M) metallic Ni particle oriented down the (110) zone axis. The Pt nanoparticle was similar, and its FFT (Figure 3g) could be ascribed to the (112) zone axis of a metallic Pt structure. In the inset of both images, a filtered image showing the (111) lattice spacing was included.

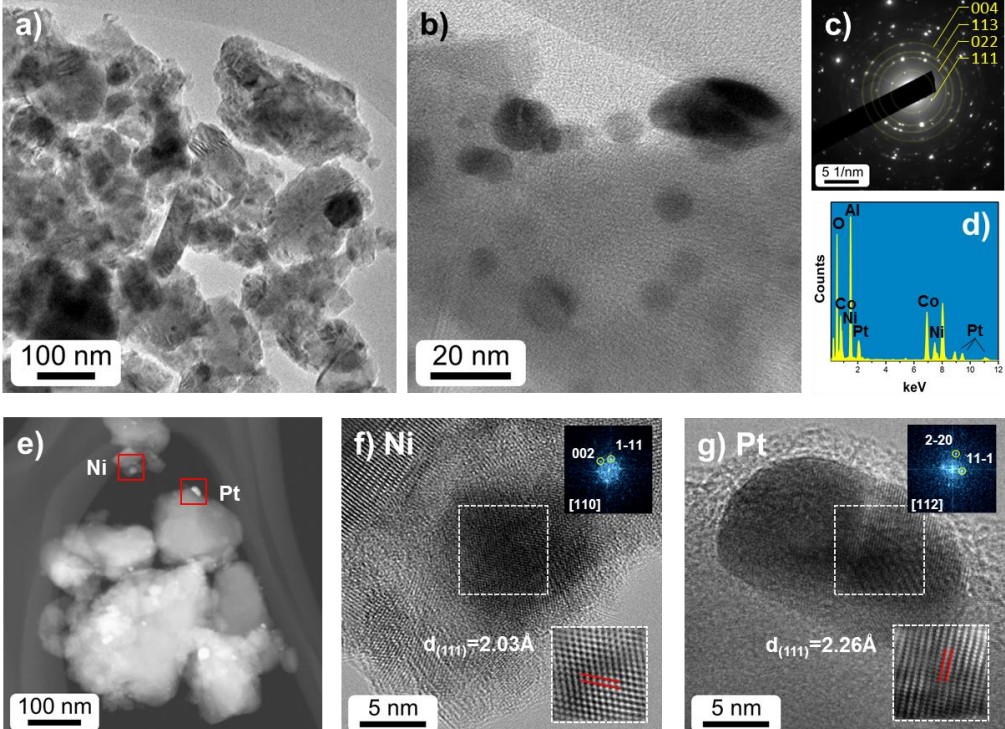

**Figure 3.** Microstructural analysis of $Pt,Ni/CoAl_2O_4$ catalyst. (**a,b**) TEM images. (**c**) SAED (Selected Area Electron Diffraction) pattern and (**d**) EDX spectrum, both corresponding to micrograph b. (**e**) HAADF-STEM (High Angle Annular Dark Field- Scanning Transmission Electron Microscopy) images. (**f,g**) Magnified micrographs of Ni (**f**) and Pt (**g**) nanoparticles; the insets correspond to the marked area of interest: (top) filtered images of the lattice fringes, and (bottom) FFT (Fast Fourier Transform) used to identify the crystallographic planes and interplanar distances.

## 2.2. Catalytic Activity

The catalytic soot combustion activity results of the temperature-programmed experiments with $NO_x/O_2$ are shown in Figure 4 and Table 1. Soot conversion was calculated as the molar fraction of total carbon in the sample released as CO or $CO_2$ in the outlet; $CO_2$ selectivity ($S_{CO2}$) is the $CO_2$ to CO + $CO_2$ molar fraction; and $NO_2$ % is the $NO_2$ to $NO_x$ molar fraction. $T_{10}$, $T_{50}$, and $T_{90}$ are the temperatures required for 10%, 50%, and 90% soot conversion, respectively.

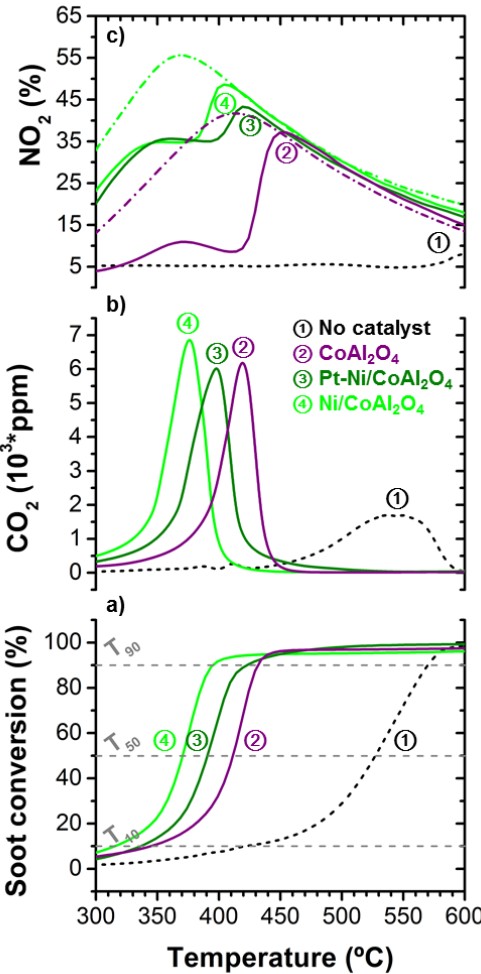

**Figure 4.** Catalytic oxidation tests: (**a**) Soot conversion profiles, (**b**) $CO_2$ concentration profiles, and (**c**) $NO_2$ profiles as $NO_2$ to $NO_x$ fraction. Reaction conditions: 100 mg catalyst + 10 mg soot, loose contact, 2500 ppm NO + 10 vol. % $O_2$ + $N_2$, 300 mL/min, 5 °C/min. Solid lines are experiments with soot and catalyst, the dotted line is the reference experiment of non-catalyzed soot combustion, and the dotted-dashed lines in (c) are blank NO oxidation experiments with catalyst but without soot.

**Table 1.** Comparison of diesel soot combustion catalysts tested in laboratory conditions ("loose" soot-catalyst contact and $NO_x$ + $O_2$ mixture).

| Catalyst | Metal Content (%) | $S_{BET}$ (m²/g) | $T_{50}$-Fresh (°C) | $T_{50}$-Reused (°C) | $S_{CO2}$ (%) | Reference |
|---|---|---|---|---|---|---|
| **No catalyst** | - | - | 527 | - | 67 | Current study |
| **CoAl₂O₄** | - | 22.9 | 411 | 494 (750 °C) 441 (500 °C) | 100 | Current study |
| **Pt,Ni/CoAl₂O₄** | 0.75% Pt 0.25% Ni | 19.3 | 390 | 421 | 100 | Current study |
| **Ni/CoAl₂O₄** | 1% Ni | 22.8 | 371 | 506 | 100 | Current study |
| **Pt/α-Al₂O₃** | 1% Pt | 15 | 398 | 422 | 100 | Current study |
| **Pt/γ-Al₂O₃** | 1% Pt | 160 | 474 | - | 100 | [58] |
| **Cu/ZnAl₂O₄** | 5% Cu | 111 | 600 | - | 95 | [59] |
| **Pt-Pd/3DOM-TiO₂** | 2% Pt-Pd | 60 | 338 | - | 99 | [60] |
| **Cu₁.₅Mn₁.₅O₄** | - | - | 397 | - | - | [61] |
| **CoCr₂O₄** | - | 59 | 396 | - | - | [62] |

Table 1 compiles the values of specific surface area, $T_{50}$, and $S_{CO2}$ of $CoAl_2O_4$, $Ni/CoAl_2O_4$, $Pt,Ni/CoAl_2O_4$, and $Pt/\alpha$-$Al_2O_3$ catalysts, along with bibliographic data for other spinel-type catalysts or supported Pt catalysts with the same platinum loading. The $CoAl_2O_4$-supported Ni and Ni-Pt

samples were more active than the barely activated cobalt aluminate spinel and the reference $Pt/\alpha\text{-}Al_2O_3$, and in the literature, under similar operating conditions, only a bimetallic $Pt\text{-}Pd(2\%)/3DOM\text{-}TiO_2$ catalyst, with double amount of noble metal, has a better performance. The sample without noble metal, $Ni/CoAl_2O_4$, had the best $T_{50}$, 156 °C below that of the uncatalyzed reaction, 27 °C lower than the $T_{50}$ of $\alpha\text{-}Al_2O_3$–supported Pt, and similar to the values obtained by using other catalysts with spinel structure as $Cu_{1.5}Mn_{1.5}O_4$ [61] and $CoCr_2O_4$ [62]. In addition, the selectivity of $Ni/CoAl_2O_4$ towards $CO_2$ formation as a soot combustion product was near 100%. In summary, the performance of the mechanically dispersed $Ni/CoAl_2O_4$ sample could be compared to that of the state-of-the-art spinel and supported metal-catalysts reported in the literature that have been tested under similar experimental conditions.

In order to better understand the catalytic behavior of the samples, the $NO_2$ to $NO_x$ fraction profiles were measured with and without soot (Figure 4c). It is recognized that the promotion of NO to $NO_2$ oxidation to conversion values close to the limits of the thermodynamic equilibrium is a key parameter in soot combustion because $NO_2$ is a much stronger oxidant than $O_2$. The bare $CoAl_2O_4$ catalyzed the oxidation of NO to $NO_2$, which was further accelerated by the presence of metal; the $Ni/CoAl_2O_4$ catalyst produced the highest amount of $NO_2$. Moreover, the presence of platinum and/or nickel did greatly increase the concentration of reactive $NO_2$ species during the reaction with soot. Figure S4 shows that the presence of the metal did not significantly contribute to the direct oxidation mechanism, as the conversion without $NO_x$ was similar to that obtained with the bare spinel [50], and only under tight contact was it slightly different from that of the uncatalyzed reaction. The tight contact is not so representative of the real conditions in catalytic traps, but it maximizes the number of contact points, which is key for the active-oxygen mechanism of soot combustion [63], and thus allows discriminating some effects better [64]. Thus, the soot combustion capacity enhancement takes place via the indirect reaction route.

## 2.3. Stability of the Catalysts

To evaluate the reproducibility of the catalytic performance, critical for practical applications, the used samples were recycled for a second soot combustion test under identical reaction conditions, and the same procedure was followed to evaluate the stability of a reference $Pt/\alpha\text{-}Al_2O_3$ catalyst. The methodology employed to recharge the reactor with soot in loose contact caused the loss of some mass of the reused SiC and catalyst mixture that must be attributed to a loss of catalyst, as SiC particles have a bigger size and come off very easily. The weight loss was quantified, but the values obtained, included in Table S2 of the Supplementary Information, might be slightly underestimated, as some quartz fibers might be present in the unloaded material. Table 1 contains the $T_{50}$ values obtained in the first and second soot combustion tests.

As expected, the reference $Pt/\alpha\text{-}Al_2O_3$ catalyst essentially maintained its initial activity. The slight increase of $T_{50}$ value in the second soot combustion test, of 24 °C, could be attributed to the methodology employed to reload the reactor and the subsequent loss of some of the catalyst mass. Similarly, in consecutive runs with the $Pt,Ni/CoAl_2O_4$ catalyst, $\Delta T_{50}$ was 31 °C. However, the activity of the samples without Pt markedly decreased in the second soot combustion test, with a shift in $T_{50}$ of 84 °C for the $CoAl_2O_4$ sample (from 411 to 494 °C) and of 135 °C for the $Ni/CoAl_2O_4$. (For more details, see Supplementary Information Figure S5a).

Figures S5 and S6 analyzed the effect of the temperature on the samples. The specific surface area of all samples slightly decreased at temperatures higher than 600 °C (see Figure S5b), which indicated that some active sites were lost during the first catalytic test. This could explain that some deactivation was always observed, but it also might indicate that operation at temperatures lower than 600 °C might minimize this effect. The Ni-based catalyst had the highest surface values after temperature treatment. However, Thermogravimetry and Differential Thermal Analysis (TG-DTA) results obtained under air atmosphere indicated a lower weight loss for the supported $Pt,Ni/CoAl_2O_4$ catalyst than for the monometallic Ni catalyst (Figure S6), and thus the better thermal stability of the former, despite the

decrease in the surface area. This fact evidenced that the metals were in part oxidized and that the presence of Pt nanoparticles improved the thermal stability to some extent. This would explain that Pt,Ni/CoAl$_2$O$_4$ had the best soot combustion efficiency in the second use, similar to that of Pt/$\alpha$-Al$_2$O$_3$ but with less amount of the noble metal.

The stability of Pt,Ni/CoAl$_2$O$_4$ for soot oxidation in diesel exhaust streams could be attributed to lower Ni oxidation due to the Pt,Ni interaction and to the presence of oxygen vacancy defects (demonstrated by XPS analysis, Figure S7), which could serve as strong anchoring sites for metal nanoparticles. Less stable atoms have greater chemical potential and bind stronger to metal atoms [65]. For instance, it has been reported that oxygen vacancies stabilize Au atoms both on TiO$_2$ and on MgO [66,67], and that Ag nanoparticles are anchored stronger to a CeO$_2$ support than to a MgO support [68]. This may be related to the greater sintering resistance of transition metals deposited on CeO$_2$ [69]. Greater resistance to the sintering of Ni-based catalysts has also been achieved by incorporating CeO$_2$ in supports [34]. Thus, the mechanical nanodispersion of Ni-Pt nanoparticles onto the surface of the milled CoAl$_2$O$_4$ spinel represents a potential way to obtain high catalytic activity and stability while reducing the amount of platinum. Finally, it must also be considered that high temperatures may promote the CoAl$_2$O$_4$ structure's reconstruction and crystallites' sintering and growth. To evaluate the effect of the temperature reached during the catalytic tests on the performance of the mechanically activated CoAl$_2$O$_4$, the experimental protocol was modified, limiting the temperature during the first run to 500 °C, and the stability test was repeated. The conversion profiles of both runs are shown in Figure 5, together with the field-emission scanning electron micrographs of the CoAl$_2$O$_4$, fresh and after the first run up to either 500 or 750 °C. In contrast to the catalytic activity decrease of CoAl$_2$O$_4$-750 in the second test, with a shift of the T$_{50}$ from 411 to 494 °C, the reused CoAl$_2$O$_4$-500 sample maintained a high catalytic activity, with T$_{50}$ of 441 °C. This confirmed that the deactivation effect could be explained by the high temperature reached in the soot catalytic combustion experiments. The wet ball milling pretreatment of the CoAl$_2$O$_4$ support modifies the morphology substantially: the particle surface is altered by the impact energy produced during the milling-process, forming defects and highly reactive sites that improve the catalytic behavior for soot oxidation [50]. However, at 750 °C, the defects generated in the CoAl$_2$O$_4$ structure were partially reconstructed, and grain boundaries and sintering necks appeared (compare Figure 5b,d), with the subsequent loss of surface area. During the catalytic test run up to 750 °C, the CoAl$_2$O$_4$ nanoparticles sintered and formed a denser and connected structure. CoAl$_2$O$_4$-500 micrograph in Figure 5c revealed an intermediate stage, in which slight densification was also observed, but the rough nanostructured surface was kept. The fact that the maximum temperature of diesel engine exhaust gases falls in the range from 260 to 540 °C makes the mechanically activated CoAl$_2$O$_4$ highly attractive for the replacement of traditional platinum catalysts.

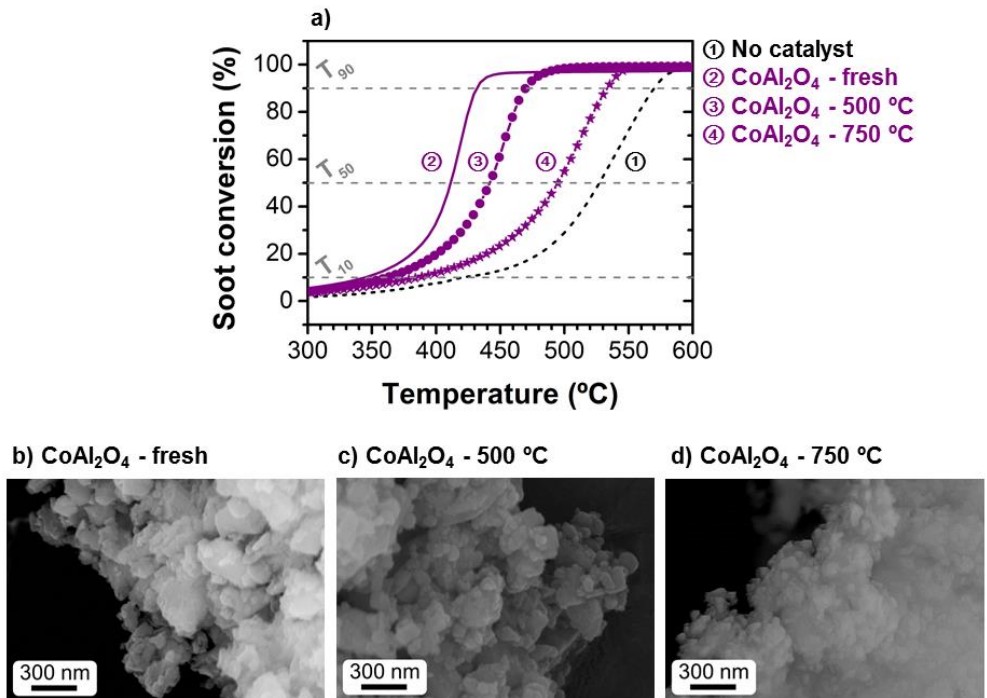

**Figure 5.** (**a**) Soot combustion profiles of CoAl$_2$O$_4$ sample—fresh (solid line) and after first use for soot combustion up to 500 °C (circles) or 750 °C (stars). The dotted line is the reference non-catalyzed soot combustion experiment. Bottom (**b–d**): FESEM micrographs of CoAl$_2$O$_4$ (**b**) fresh sample, and reused after (**c**) 500 °C or (**d**) 750 °C.

## 3. Materials and Methods

### 3.1. Catalysts Preparation

As-received CoAl$_2$O$_4$ spinel microparticles with an average particle size of ~2 μm (Manuel Riesgo, S.A., Madrid, Spain) were mechanically pre-activated by ball milling in 33% ethanol medium with 1 mm cerium-stabilized zirconia balls (balls to powder mass ratio equal to 1) in a polyamide type jar (Spex 8000 Mixer mill milling equipment, Madrid, Spain) at 50 rpm for 3 h. Pt,Ni/CoAl$_2$O$_4$ with 0.75 wt. % Pt and 0.25 wt. % Ni, and Ni/CoAl$_2$O$_4$ with 1 wt. % Ni were prepared by mechanical milling in ethanol to incorporate the appropriate amount of high purity (>99.9%) PtO$_2$ and/or NiO nanoparticles (Sigma-Aldrich, Madrid, Spain) onto the micrometric CoAl$_2$O$_4$ support. As-received PtO$_2$ and NiO nanopowders, with an average particle size of ~20 nm, formed large agglomerates with diameter >20 μm and >10 μm, respectively, so they were pre-dispersed in a 60 cm$^3$ nylon container for 1 h at 500 rpm. The process was assisted by the addition of zirconia balls (1 mm in diameter) and 20 cm$^3$ of ethanol. The balls were cleaned before the milling with a typical mixture of 3 parts of concentrated sulfuric acid (H$_2$SO$_4$) and 1 part of 30% hydrogen peroxide solution (H$_2$O$_2$). This solution is highly corrosive and an extremely powerful oxidizer, and, consequently, treated surfaces must be reasonably clean and completely free of residues. Then, the activated CoAl$_2$O$_4$ microparticles were added, and the mixture was homogenized for 10 min. Finally, the sample was dried and thermally treated at 550 °C under N$_2$–H$_2$ atmosphere to reduce the oxides to the metallic phase and remove all possible nitrate species from the platinum nitrate [70]. A reference Pt (1%)/α-Al$_2$O$_3$ catalyst was prepared by a similar low-energy wet milling procedure from PtO$_2$ nanoparticles and micrometric α-Al$_2$O$_3$, as described elsewhere [52]. α-Al$_2$O$_3$ was used instead of typical porous γ-Al$_2$O$_3$ support to favor the mechanical nanodispersion process and because this material has demonstrated a comparable efficiency to similar materials impregnated on γ-Al$_2$O$_3$.

## 3.2. Catalysts Characterization

The crystalline phases were characterized by X-ray diffraction (XRD, D8, Bruker, Madrid, Spain), using a Lynx Eye detector and Cu K$\alpha_1$ radiation.

The Brunauer–Emmett–Teller (BET) equation was used to determine the surface area of the catalysts from $N_2$ adsorption-desorption data obtained in a Micrometrics ASAP2020 analyzer (Madrid, Spain).

The morphology of the powders was evaluated by using secondary electron images of field emission scanning electron microscopy (FE-SEM, Hitachi S-4700, Madrid, Spain).

The particle size and morphology at the nanoscale were also evaluated using a transmission electron microscope (TEM/HRTEM, JEOL 2100F, Madrid, Spain) operating at 200 kV and equipped with a field emission electron gun, providing a point resolution of 0.19 nm. HRTEM filtering and image processing were performed using Digital Micrograph software.

The surface composition was determined using an X-ray Photoelectron Spectrometer (XPS, K-Alpha, Thermo Scientific, Madrid, Spain), equipped with a monochromated Al K$\alpha$ (1486.6 eV) source running at a voltage of 12 KV. A pass energy of 200 eV was used for survey scans, while, for high-resolution scans, the pass energy was 40 eV. Finally, for charge correction, a 1-point scale with the C 1s peak shifted to 285 eV was used.

Differential Thermal Analysis (DTA) and Thermogravimetric (TG) analysis were carried out from 20 to 1000 °C with a heating rate of 5 °C/min in a Netzch STA 409 Thermo-Analyzer (Madrid, Spain).

## 3.3. Catalytic Activity Evaluation

Temperature Programmed Oxidation (TPO) of soot was studied under NO + $O_2$ or air atmospheres. TPO tests under $NO/O_2$ were conducted in a fixed-bed tubular microreactor ($\phi$ = 4 mm; bed length = 70 mm) from 25 to 750 °C with a heating rate of 5 °C/min. 100 mg of catalyst was mixed with 10 mg of soot particles obtained by diesel fuel combustion [71] and stirred with a spatula for 5 min in order to reproduce the loose contact conditions [72]. The mixture was diluted in 1000 mg of SiC (particle size 500 μm), to prevent the formation of hot spots in the catalytic bed, and placed in the center of the tubular microreactor, in contact with a k-type thermocouple used to control the catalyst bed temperature. In a typical test, a gas stream of 300 mL/min containing 2500 ppm $NO_x$ in 10% $O_2/N_2$ was passed through the catalyst-soot diluted mixture at a Gas Hourly Space Velocity (GHSV) of 150,000 h$^{-1}$. Reference experiments with only the catalyst or with only soot were also conducted. The reactor outlet was analyzed in a Thermo Nicolet Fourier transform infrared (FT-IR) spectrometer (Madrid, Spain) equipped with a gas cell heated at 120 °C.

The stability of the catalysts was tested by submitting them to a second combustion test under the same reaction conditions. After the first test, the reactor was cooled down to room temperature, and the mixture of SiC and catalyst was carefully unloaded and recharged with the same amount of soot as in the previous test, stirring with a spatula for 5 min in order to reproduce the loose contact conditions.

TPO tests with only $O_2$ as oxidant were performed under loose and tight contact conditions in a simultaneous thermal analyzer (STA 6000) connected to a Frontier FTIR spectrometer equipped with a heated gas cell, both from PerkinElmer (Madrid, Spain). To obtain tight contact, catalyst and soot were mixed in a mortar. Around 25 mg of powder was placed in alumina crucibles and subjected to a ramp of 10 °C min$^{-1}$ up to 950 °C in air. Gas-phase IR spectra were collected from 650 to 4000 cm$^{-1}$ at a resolution of 2 cm$^{-1}$ with 2 accumulations.

## 4. Conclusions

$CoAl_2O_4$ activation and Pt and/or Ni nanodispersion onto $CoAl_2O_4$ could be successfully achieved using a mechanical method, which was not previously used to obtain these specific materials. These catalysts showed high efficiency for soot combustion in the presence of NO, despite the relatively big size of the metal particles obtained in the low specific surface area of the spinel, due to the easier soot-to-catalyst contact and the efficient generation of oxidant $NO_2$ by the system. The stability of

the activated $CoAl_2O_4$ was maintained in the range of temperatures of diesel engine exhausts, but care must be taken to avoid hot spots formation, for example, in regions where large soot aggregates burn, as this would lead to thermal deactivation. The mechanically activated $CoAl_2O_4$ spinel is highly attractive as a catalyst and as support for the replacement of traditional platinum catalysts. The highest reaction rates were observed for the monometallic samples prepared with 1 wt. % Ni ($Ni/CoAl_2O_4$), but this high performance was not maintained in successive uses. On the contrary, the substitution by Pt of 1/4 of the Ni loading seemed to hinder the metal oxidation, and the bimetallic Pt,Ni nanoparticles dispersed onto mechanically activated $CoAl_2O_4$ showed good stability. Therefore, these catalysts are promising candidates for oxidation of diesel soot particles because of their easy synthesis, high activity, and low cost.

**Supplementary Materials:** The following are available online at http://www.mdpi.com/2073-4344/10/4/406/s1, Figure S1: XRD patterns of the mechanically activated support and the supported catalysts, and JCPD standards for $CoAl_2O_4$, α-$Al_2O_3$, Pt, and Ni. Figure S2: (**a**,**c**,**e**) Survey XPS spectra. (**b**,**d**,**f**) XPS spectra of Co 2p core levels. Figure S3: (**a**) HAADF-STEM image of $Pt,Ni/CoAl_2O_4$ catalyst. (**b**,**c**) EDX mapping analyses for Ni (green) and Pt (gray) of the area. Figure S4: Soot combustion with $O_2$ (in the absence of NO) without catalyst and over the $Pt,Ni/CoAl_2O_4$ and $Ni/CoAl_2O_4$ catalysts under tight and loose contact conditions. The temperature of maximum $CO_2$ production is indicated. Figure S5: Stability evaluation. (**a**) The temperature needed for 50% soot combustion under loose contact between catalyst and soot with $Ni/CoAl_2O_4$ and $Pt,Ni/CoAl_2O_4$ catalysts, $CoAl_2O_4$ support, and $Pt/Al_2O_3$ reference. Solid: first use, dashed: second use. (**b**) Effect of the temperature on the BET specific surface area of the prepared samples. Solid: BET measured before thermal treatment, dashed: BET measured after thermal treatment of samples at 800 °C. (**c**,**d**) Thermal analysis in the air of the reduced catalysts (c) $Ni/CoAl_2O_4$ and (d) $Pt,Ni/CoAl_2O_4$. Figure S6: Thermogravimetric behavior under air atmosphere of the fresh samples: (**a**) $Pt/Al_2O_3$, (**b**) $CoAl_2O_4$, (**c**) $Pt,Ni/CoAl_2O_4$, (**d**) $Ni/CoAl_2O_4$, (**e**) $Ni(5\%)/CoAl_2O_4$. Figure S7: O 1s XPS spectra of the catalyst samples. Table S1: Elemental composition (at %) of the samples based on XPS analysis (traces of Zr not quantified). Table S2: Quantification of the weight loss generated during the methodology employed to recharge the reactor.

**Author Contributions:** C.M.Á.-D.: Methodology, Formal analysis, Validation, Investigation, Writing—Original Draft, Visualization. R.P.: Conceptualization, Methodology, Validation, Resources, Writing—Review and Editing, Supervision, Project Administration. J.J.R.: Conceptualization, Writing—Review and Editing, Formal Analysis. F.R.-M.: Software, Formal Analysis, Data Curation. L.P.: Software, Formal Analysis, Data Curation. J.F.F.: Conceptualization, Resources, Writing—Review and Editing, Supervision, Project Administration, Funding Acquisition. All authors have read and agreed to the published version of the manuscript.

**Funding:** This research was funded by CSIC, NANOMIND project number CSIC201560E068, and the Spanish Government, grant number MAT2017-86450-C4-1-R projects. C.M.A.-D. received financial support from the Spanish Ministry of Economy, Industry and Competitiveness (MINECO) for an FPI grant BES-2014-069779, which is co-financed with FEDER funds. F.R.-M. is indebted to MINECO for a 'Ramon y Cajal' contract (ref: RyC-2015-18626), which is co-financed by the European Social Fund. F.R.-M. also acknowledges support from a 2018 Leonardo Grant for Researchers and Cultural Creators (BBVA Foundation).

**Conflicts of Interest:** The authors declare no conflicts of interest.

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
