# Peer review of "Performance and Stability of Wet-Milled CoAl2O4, Ni/CoAl2O4, and Pt,Ni/CoAl2O4 for Soot Combustion"

_catalysts, doi:10.3390/catal10040406_

Round 1

Reviewer 1 Report

Dear authors,

presented paper well done and fine structured. I can recommend it to be published. The only wish is to exclude figure 1 from the main text and add it to supplementary materials.

Reviewer 2 Report

The authors present a report on the catalysts for soot combustion from the diesel exhaust gasses. The materials are thoroughly characterized and complementary catalytic tests are performed.

Unfortunately, the authors fail to deliver an insight into the modes of action of the presented catalytic material Ni-Pt/CoAl2O4.

  • no mention of manganese oxides with alkali, and other metal oxide-based materials as soot combustion catalysts in the introduction;
  • specify, how the balls were cleaned before the milling, and whether the contamination of the samples with Zr (Ce) can be avoided;
  • XPS analysis does not provide information about material's structure, therefore it is not justified to infer the location of cobalt in the CoAl2O4 samples;
  • the EDX analysis shows 65-35 % Pt-Ni, while the target was 25-75 % Pt-Ni - it is hardly in the range of theoretical formulation;
  • P7.L210  - what does it mean in the structure?
  • there is no quantification of the acid sites to support the claim about decrease of the CoAl2O4 acidity;
  • P10.L292 - the method is not novel- the application of this method to the specific catalytic material for the specific reaction may be;

Most important issues:

  • the authors are focusing on the effects of the high temperature on the support CoAl2O4 with less attention for the active phase;
  • there is no proof of Ni-Pt nanoparticles - in fact, the authors show distinct Ni and Pt particles in TEM figures;
  • what is the nature of interaction of Ni and Pt in the studied sample?
  • the chosen reference sample is does no allow for fair comparison; as authors stated Al2O3 do not appreciably convert NO to NO2 - without a Pt/CoAl2O4 sample any meaningful conclusions could not be drawn;
  • Ni could enter the spinel phase - accounting for the determined Ni/Pt ratio by EDX, and the reactivity - have the authors considered that?
  • no optimization of Pt content - different ratios of Ni/Pt would help understanding the effect of Pt
  • there is no indication what is the role of Pt - maybe it takes over most of the catalytic action in place of Ni after tests;

Reviewer 3 Report

The reviewer has a minor query regarding the manuscript, which is given below.

(1) Page 6, line 196-198: Ni/CoAl2O4 sample has the best T50, 27oC lower than the T50 of Pt/α-Al2O3 catalyst in Table 1. However, the authors made a mistake in this temperature difference (8oC).

=> (Table 1) T50-fresh: Ni/CoAl2O4 (371 oC), Pt/α-Al2O3 (398 oC)

Round 2

Reviewer 2 Report

The authors made reasonable corrections to their manuscript . I recommend publication.